# Ibrutinib in Gynecological Malignancies and Breast Cancer: A Systematic Review

**DOI:** 10.3390/ijms21114154

**Published:** 2020-06-10

**Authors:** Julian Matthias Metzler, Laurin Burla, Daniel Fink, Patrick Imesch

**Affiliations:** Department of Gynecology, University Hospital Zurich, 8091 Zurich, Switzerland; laurin.burla@usz.ch (L.B.); daniel.fink@usz.ch (D.F.); patrick.imesch@usz.ch (P.I.)

**Keywords:** ibrutinib, Bruton’s tyrosine kinase, Btk, kinase inhibitor, ovarian cancer, endometrial cancer, breast cancer, gynecology, gynecologic oncology, solid tumors

## Abstract

Ibrutinib is an orally available, small-molecule tyrosine kinase inhibitor. Its main purpose is to inhibit Bruton’s tyrosine kinase (BTK), an enzyme that is crucial in B cell development. It is FDA approved for the treatment of certain hematological malignancies. Several promising off-target drug effects have led to multiple, mostly preclinical investigations regarding its use in solid tumors. Unfortunately, data on its effectiveness in gynecological malignancies are limited, and (systematic) reviews are missing. The objective of this review was to summarize the existing literature and to analyze the evidence of ibrutinib as a treatment option in gynecological malignancies, including breast cancer. Studies were identified in MEDLINE and EMBASE using a defined search strategy, and preclinical or clinical research projects investigating ibrutinib in connection with these malignancies were considered eligible for inclusion. Our findings showed that preclinical studies generally confirm ibrutinib’s efficacy in cell lines and animal models of ovarian, breast, and endometrial cancer. Ibrutinib exerts multiple antineoplastic effects, such as on-target BTK inhibition, off-target kinase inhibition, and immunomodulation by interference with myeloid-derived suppressor cells (MDSCs), programmed death-ligand 1 (PD-L1), and T cell response. These mechanisms were elaborated and discussed in the context of the evidence available. Further research is needed in order to transfer the preclinical results to a broader clinical appliance.

## 1. Introduction

This publication offered a systematic review of ibrutinib’s current use in gynecological malignancies. The article is structured in three parts. After a conceptual and terminological introduction to the drug and the diseases, genes, and proteins covered (I); we next reviewed the existing literature on ibrutinib’s use in gynecological malignancies, including breast cancer (II); Finally, we analyzed the results and discussed ibrutinib’s multiple mechanisms of action in the context of the literature available (III).

Ibrutinib is an orally available, small-molecule tyrosine kinase inhibitor. Its main purpose is to inhibit Bruton’s tyrosine kinase (BTK), a crucial enzyme in the B cell antigen receptor signaling pathway [1,2,3]. BTK is a member of the TEC-family kinases (TEC, BTK, BMX, ITK, RLK), a group of non-receptor kinases that modulate intracellular signaling in B- and T-cells and affect cell proliferation, survival, and differentiation in various cell types [4]. Ibrutinib blocks BTK by forming covalent bonds with cysteine 481 near the ATP binding pocket of the enzyme, resulting in an irreversible inactivation and suppression of downstream effectors [2,4]. In this setting, ibrutinib is FDA approved for the treatment of adult patients with hematological diseases, such as mantle cell lymphoma (MCL), chronic lymphocytic leukemia (CLL), small lymphocytic lymphoma (SLL), Waldenstrom’s macroglobulinemia (WM), marginal zone lymphoma (MZL), and chronic graft versus host disease (cGVHD) [5].

Besides its well-known and approved effectiveness in hematology, ibrutinib shows some promising off-target potential due to its efficacy on additional binding sites and enzymes, including tyrosine kinases that conserve the same Cys481 residue, such as the ERBB family (EGFR, ERBB2/HER2, et al.) [2]. This has led to several, mostly, preclinical publications, investigating its use in solid tumors [4,6,7,8]. With an increasing number of research papers addressing this topic, and new kinase inhibitors (such as neratinib) being legally approved in this field, the effectiveness of ibrutinib in the treatment of gynecological malignancies and breast cancer seems plausible, although summarized evidence is missing.

Ibrutinib’s toxicity is mainly explained by its inhibition of related kinases. EGFR inhibition may cause rash and diarrhea. Its more serious unwanted effects include cardiac events (atrial fibrillation, rarely ventricular arrhythmia) and bleeding, as BTK and TEC modulate platelet aggregation [9,10,11]. Further and frequently described side effects include digestive and myelosuppressive disorders, hypertension, edema, fatigue, and infections [4]. The side effects are generally well manageable, and combinations of ibrutinib with multiple chemotherapeutic agents and monoclonal antibodies are safe [3,12,13,14]. The dosage depends on the underlying disease and ranges from 420 mg daily in CLL/WM to 560 in MCL. In current trials on solid tumors, daily doses from 420 to 840 mg are being administered [15,16,17]. Dose reductions are recommended after certain recurrent toxicities ≥ CTCAE 3°.

The nonspecificity of ibrutinib has led to the development of several second-generation irreversible BTK inhibitors [18]. Acalabrutinib is FDA approved for adults with MCL, CLL, and SLL [19]. It inherits a higher sensitivity to BTK, while it has little or no inhibitory effect on other kinases, such as ITK, EGFR, ERBB2, ERBB4, JAK3, BLK, FGR, HCK, LCK, SRC, and YES1 [20,21]. Further compounds include zanubrutinib (FDA approved for the treatment of MCL [22]), tirabrutinib, spebrutinib, and evobrutinib. All of these drugs show a different selectivity profile against off-target enzymes and are mainly investigated in B cell malignancies or autoimmune diseases [23,24,25]. While the reduced off-target activities of these second-generation BTK inhibitors are thought to minimize side effects [9], this might arguably also minimize their effect on solid malignancies when compared to ibrutinib. As shown below, many of ibrutinib’s antineoplastic mechanisms in solid tumors rely on its kinase promiscuity with efficacy on enzymes, such as EGFR and ERBB2. Furthermore, data on the efficacy of these newer compounds on diseases outside of the hematological spectrum are scarce or missing. Hence, this review focused on publications investigating ibrutinib.

Ovarian cancer is one of the leading causes of death from gynecological cancers in developed countries, with a 5-year survival rate of less than 50% after initial diagnosis. First-line therapy includes a combination of surgical tumor resection and platinum-based chemotherapy [26]. Current guidelines recommend the subsequent use of maintenance therapy with a poly-ADP-ribose-polymerase (PARP) inhibitor in a selected group of patients with stage II–IV disease and BRCA1/2 mutations, who are in complete or partial remission after primary surgery and chemotherapy [26]. Therapy response is commonly monitored by measuring levels of CA-125, which is also used to detect residual or recurrent disease, as well as clinical decision-making regarding imaging and second-look procedures [27].

In cases of recurrent disease, a distinction is made between platinum-sensitive and platinum-resistant recurrence. The classical definition of platinum sensitivity is recurrence after a platinum-free interval of ⩾ 6 months. There has been some recent controversy regarding this old definition, questioning whether a definition based on the response to platinum would be more biologically relevant [28,29].

In cases of platinum-sensitive recurrence, a similar approach of cytoreductive surgery followed by platinum-based chemotherapy is used. Platinum-resistance is typically seen after prolonged use of platinum-based chemotherapy due to the development of platinum-resistant clones. In this situation, platinum-free cytotoxic regimens, targeted agents, and/or best supportive care are recommended in cases of radiological or clinical relapse [30]. In cases with a purely “biochemical relapse” (increasing CA-125 without radiologic or clinical evidence of the disease), treatment can be delayed until the occurrence of symptoms [30].

Breast cancer is the most common cancer in the female population, accounting for almost a third of all newly diagnosed cancers in women. Despite great advancements in therapy, it remains the second leading cause of cancer-related deaths in women due to its high incidence [31].

Breast cancer is typically categorized into subtypes, depending on the expression of hormone receptors (estrogen, progesterone) and of human epidermal growth factor 2 (HER2/ERBB2). The three major subtypes, namely, hormone-receptor-positive/HER2-negative cancers (70%), HER2-positive cancers (15–20%), and triple-negative cancers (15%), vary in prognosis but offer distinctive options for targeted or systemic treatment [32].

Over 90% of patients present with the non-metastatic disease at diagnosis and are treated with curative intent. This usually consists of a local (surgical tumor resection) and a systemic therapy according to the molecular subtype, including endocrine treatment, antibodies against HER2, and chemotherapy, with a trend towards neoadjuvant systemic treatment in recent years. Adjuvant radiotherapy is usually recommended after breast-conserving surgery [32].

In cases of metastatic disease, treatment is palliative and depends on the subtype. Therapeutic choices include endocrine treatment and cyclin-dependent kinase inhibitors for hormone-receptor-positive subtypes, HER2-targeted antibodies, antibody compounds or tyrosine kinase inhibitors for HER2-positive tumors, and chemotherapy. Radiotherapy can be used for local disease control. Recently, the availability of immunotherapy targeting PD-1 and PD-L1 has led to promising effects in HER2^+^ and triple-negative breast cancer.

## 2. Results

### 2.1. Overview

Of the 92 search results, 12 articles were excluded due to the selection criteria or duplication. After the abstract/full-text assessment, 69 were excluded because they were unrelated to the topic. Eleven publications were selected as eligible for inclusion. Figure 1 shows the flow diagram for the identification of relevant studies. In this results section, the evidence available on the efficacy of ibrutinib in breast cancer and gynecological malignancy, as summarized in Table 1, Table 2 and Table 3, has been reviewed.

### 2.2. Review of Identified Studies

#### 2.2.1. Breast Cancer

In 2014, **Grabinski and Ewald** [33] tested the efficacy of ibrutinib (alone or in combination with dactolisib, a PI3K/mTOR inhibitor) on ten different subtypes of breast cancer cells in vitro. Cell viability after treatment with different concentrations of ibrutinib was measured. While ER+ and triple-negative cell lines were not significantly attenuated after drug treatment, a significant reduction in the viability of HER2^+^ cell lines was demonstrated, even with low drug concentrations (half-maximal inhibitory concentration IC50 in the lower nanomolar range). Specific inhibition of the phosphorylation of the cancer-related signaling pathway AKT in HER2^+^ cells was demonstrated and was postulated as the underlying mechanism of action. The addition of the dual PI3K/mTOR inhibitor dactolisib further reduced cell viability in a synergistic manner. Moreover, ibrutinib reduced the phosphorylation and downstream signaling of the EGFR receptors—ErbB1, ErbB2, and ErbB3. In summary, the study suggested a potential use of ibrutinib in HER2^+^ breast cancer.

**Sagiv-Barfi** et al. [34] investigated the synergistic effects of ibrutinib in combination with anti-PD-L1 antibodies on different cell lines, including 4t1-Luc, a triple-negative breast cancer (TNBC) cell line. As 4T1-cells express low levels of PD-L1 only and no BTK, no single drug showed any effect on tumor viability in vitro or in 4T1-xenografted mice. However, a reduction in tumor size, fewer lung metastases, and an increased survival along with the generation of specific antitumor T-cells were observed with a combined treatment of ibrutinib and anti-PD-L1.

Given the fact that ibrutinib may have relevant HER2 targeting activity, **Chen et al.** [2] sought to address several critical questions regarding its potential clinical use in HER2-overexpressing breast cancer through growth inhibition experiments, apoptosis assays, xenograft mouse models, pharmacokinetics/pharmacodynamics, immunohistochemical analysis, Western blot analysis, cell cycle assay, BTK occupancy assays, and kinase assays. In these experiments, they were able to show that ibrutinib demonstrated antiproliferative activity against certain solid tumor cell lines, similar to the previously published patterns of the known EGFR inhibitors—gefitinib and lapatinib. Cell lines that depend on HER2 for their proliferation were most sensitive. Mechanistically, the compound was found to inhibit phosphorylation and downstream signaling of HER2 and EGFR in these cell lines at concentrations in the nanomolar range, mirroring the effect of other HER2 inhibitors. In comparison to the potency of other ERBB kinase inhibitors, ibrutinib had a higher potency than lapatinib and gefitinib, albeit lower than afatinib and neratinib.

In a series of xenograft studies with HER2^+^ cell lines in mice, ibrutinib slowed tumor progress up to 90% in a dose-dependent manner in certain series. The maximum effect was seen in mice strains that achieved a higher drug exposure (probably due to differences in bioavailability), as revealed by pharmacokinetics. As a promising outlook, the calculated AUC (area under the plasma drug concentration-time curve) for tumor growth inhibition seemed achievable in humans with a standard 560 mg per day dosing.

**Stiff et al**. investigated the effect of ibrutinib on myeloid-derived suppressor cells (MDSCs) [35]. In tumor-bearing hosts, these cells increased in response to factors released by tumors and stromal cells, and elevated expression subsequently affected immune response and diminishes the efficacy of immunotherapy. In their work, they were able to document that human and murine MDSCs expressed BTK. Furthermore, ibrutinib inhibited the formation of human MDSCs in vitro. In the next experiment, ibrutinib was fed to mice with xenografted triple-negative mammary tumor cells. This led to a reduction in immune-suppressing MDSCs in the murine spleen and tumor. Lastly, a murine breast cancer model was used to assess the efficacy of a combined ibrutinib/anti-PD-L1 treatment, resulting in a significant enhancement of the immunotherapies’ efficacy.

After reporting that an isoform of BTK (BTK-C) protects breast cancer cells from apoptosis in 2013 [36], in 2016, **Wang et al.** showed that inhibition of BTK by ibrutinib could decrease cancer cell survival and prevent drug resistance [37]. They found that HER2^+^ cell lines were more sensitive to drug treatment than luminal, triple-negative, or even nontumorigenic cell lines in cell growth assays. The effect of ibrutinib on monolayer and 3D culture was even more potent than that of lapatinib. As described earlier [33], the phosphorylation of EGFR, HER2, Her3, and ErbB4 was inhibited, resulting in a blockage of activating downstream pathways. These effects were confirmed in vivo, together with growth inhibition of the xenografted tumors in mice.

The authors emphasized ibrutinib’s dual effect on two key growth and survival kinases, namely, BTK and the EGFR family. This might explain its greater impact on HER2^+^ cells and provide a strong rationale for its clinical use, especially since a correlation between the expression of HER2 and BTK was found in human breast cancer tissue.

As the development of new cancer drugs is slow and costly, **Di et al.** [38] generated a novel computational approach in order to determine candidate drugs that could possibly be repurposed to treat cancer. By using known drug-indication correlations and combining them with drug-drug and indication-indication similarities, they were subsequently able to hypothesize new drug-indication associations.

The software used raw data from the National Cancer Institute (NCI)-60 cancer cell line database, a panel containing extensive data on the gene expression profiles and pharmacology for 60 cancer cell lines, including breast and ovarian cancer. These data were used to study the correlation between mRNA/microRNA expression levels and drug susceptibility. They created a drug-drug functional similarity network in order to predict drug functional similarities at a biological pathway level. The system was validated using cross-validation tests on approved drugs, receiver-operating characteristic curve analysis, and literature searches, confirming its efficiency in prioritizing candidate cancer drugs.

Sixteen FDA approved drugs for the treatment of breast cancer were mapped to the described network. The algorithm then identified 14 candidate drugs that were statistically significant. In this list of prioritized scores, ibrutinib ranked 3rd out of more than 3000 substances tested and was “considered to show great potential therapeutic effects” [38]. Moreover, ranks 1 and 2 (gefitinib and afatinib) represented two EGFR inhibitors, additional proof of the importance of this receptor in breast cancer treatment. After Sagiv-Barfi et al. showed a preclinical effect of ibrutinib and anti-PD-L1 antibodies in lymphoma, breast cancer, and colon cancer [34], **Hong et al**. initiated an open-label, multicenter, phase Ib/II study evaluating ibrutinib plus durvalumab in humans [39]. Inclusion criteria included patients with stage III/IV breast cancer, pancreatic adenocarcinoma, and non-small-cell lung cancer (NSCLC). After defining a recommended phase 2 dose of 560 mg ibrutinib p.o. daily and durvalumab 10 mg/kg intravenously (i.v.) twice a week in phase 1, they continued to treat 122 patients at that dose. Of these patients, 45 had breast cancer (28 TNBC, 17 HER2^+^), and all of them had previously been treated with ≥ 2 prior therapies. Ninety-four percent of the patients had stage IV disease at a median age of 61 years. An overall response rate of 3% was found in patients with breast cancer, with a median progression-free survival rate of 1.7 months. Pancreatic cancer and NSCLC ranged at an overall response rate of 2% and 0%, with a median progression-free survival rate of 1.7 and 2.0 months, respectively. While the safety profiles of both drugs were consistent with previously known data and were considered acceptable, the antitumor activity in this pretreated, late-stage study population was limited.

**Varikuti et al**. [40] used an orthotopic mouse breast cancer model to investigate ibrutinib’s effect on tumor progression and metastasis. Mice were injected with tumor cells resembling a luminal b subtype [41]. After a growth period of one week, mice were randomized to treatment or placebo groups. Tumor progression, as well as tumor weight, were significantly reduced in the treatment group. BTK expression in tumor tissue was decreased in a dose-dependent manner. Mice receiving the verum also had fewer lung metastases and splenomegaly when compared to the placebo group. Immunosuppressive monocytic MDSCs were found to be reduced in the tumors and spleens of treated mice with an increase in mature dendritic cells (DCs). Furthermore, ibrutinib initiated a phenotype switch from MDSCs to DCs in vitro. Regarding immune response, the team was able to show that ibrutinib increased T cell effector functions, encouraged T cell proliferation, and promoted Th1-dominant cytokines in vivo. 

Table 1 shows a summary of the studies described.

#### 2.2.2. Gynecological Malignancies

**Zucha et al.** [42] studied BTK and its inhibition by ibrutinib comprehensively in human ovarian cancer tissue samples. They demonstrated a high expression of BTK in malignant cells with more intense staining in metastatic and late-stage disease.

A further experiment demonstrated that cell lines highly resistant to platinum expressed higher levels of cancer stem cell (CSC) markers than cells with moderate or absent platinum resistance, indicating a higher fraction of CSCs in these cells.

Next, spheroids from the parental cell lines were generated, and the cisplatin responsiveness of spheroids and parental cell lines were compared. Spheroids of CSC-rich cell lines showed high resistance to platinum, indicating a major role of CSCs in cisplatin resistance, while BTK was critical in regulating ovarian CSCs.

To prove the hypothesis that BTK mediates platinum resistance via regulation of CSCs, loss-of-function and gain-of-function studies were performed. In a primarily platinum-responsive cell line, the overexpression of BTK stimulated the activation of transcription factor STAT3. Consequently, several downstream effectors, known for their role in stem cell regulation and cell survival, such as SOX2 and Bcl-xL, were upregulated. Following this “stemness” gene regulation, these cells formed spheroids and increased their CSC count. In contrast, JAK2/STAT3 targets (such as Bcl-xL) were downregulated in BTK knockdown loss-of-function studies. Treatment with ibrutinib showed a similar effect, decreasing BTK phosphorylation and Sox2/Bcl-xL expression. This diminished the self-renewal capacities of highly malignant cells and their proportion of CSCs, which ultimately reduced the population and its spheroid-forming abilities. Cell lines initially highly resistant to cisplatin became more sensitive after BTK knockdown. A synergistic effect of ibrutinib and cisplatin was shown in serous and clear cell lines, indicating the possible use of ibrutinib as a platinum sensitizer in these malignancies.

There also is evidence for an antiproliferative role of ibrutinib in endometrial cancer. **Tamura et al**. [43] created patient-derived tumor organoids (PDOs) from human tumor tissue (lung, ovarian, endometrial). The morphology, histology, and gene expression of these organoids were similar to their source tumor. The reaction to a total of 61 anticancer agents was tested in cell growth inhibition experiments. Subsequently, the efficacy of each anticancer agent was evaluated. A PDO called REME9 was derived from carboplatin/paclitaxel-resistant endometrial cancer. As expected, carboplatin and paclitaxel showed only weak effects on this cell line, an observation that suggested certain comparability between the PDOs and the source tumor regarding the drug response. Ibrutinib led to growth inhibition in these cells at concentrations (AUC) notably lower than carboplatin and paclitaxel, but also lower than MTX and vinca alkaloids—substances that current guidelines recommend as a possible treatment for metastatic patients. However, ibrutinib had much less effect in clear-cell adenocarcinoma specimens. A comparison with the effect of anthracyclines was not possible, as this substance class was not investigated.

The drug-drug functional similarity network created by **Di et al.** [38] has been described in detail in in Section 2.2.1 above. Even though ibrutinib achieved a high score as a potential agent for breast cancer treatment, it failed to thrive in the experiments on ovarian cancer and ranged in the last sextile of the candidate drug list ranking.

As treatment options in platinum-resistant ovarian cancer are scarce, **Lohse et al.** [44] investigated the treatment response to 30 FDA-approved chemotherapeutics in six such patient-derived cell lines, using ex vivo drug sensitivity testing. The experiments were performed on two endometrioid, two clear-cell, and two papillary-serous cell lines, respectively. To interpret and compare the different drugs, the modified drug sensitivity score was used, a marker incorporating different parameters, such as potency, efficacy, effect range, and therapeutic index, allowing for prioritization of compounds with one sole numerical metric. In light of this *personalized medicine* approach, differing treatment responses between cell lines were observed. While docetaxel and cephalomannine diminished the survival of all cell lines tested, the effect of ibrutinib varied greatly. A weak inhibitory effect was found on one endometrioid and papillary-serous cell line, respectively, while no effect was seen on clear-cell cancer lines. Ibrutinib generally provoked less treatment response compared to the tested antimitotics, such as docetaxel. The response to each drug varied greatly and was partly unpredictable between the cell lines tested. Afatinib, for example, another kinase inhibitor tested, was the top candidate against papillary-serous cell line P5X but showed the poorest specificity against the second papillary-serous cell line P9A1.

Table 2 shows a summary of the studies investigating ibrutinib in gynecological malignancies.

### 2.3. Clinical Trials

Three trials investigating ibrutinib’s use in gynecological malignancies were identified. Figure A1 (Appendix A) shows the flow diagram for the identification of relevant trials.

NCT02403271 is a phase Ib/II trial, assessing the safety and efficacy of a combination of ibrutinib (560 mg daily) and durvalumab in patients with relapsed or refractory solid tumors. As the results have already been published by Hong et al. [39], please refer to the “Results” section above for further details. 

NCT03379428, “Trial of Ibrutinib Plus Trastuzumab in Her2-Amplified Metastatic Breast Cancer”, is a phase I/II, open-label, nonrandomized dose-escalation study. Its primary objectives are to define a maximum tolerated dose of oral ibrutinib (420, 560 or 840 mg, phase I) and to assess the clinical benefit rate (phase II) in patients with HER2^+^ metastatic breast cancer. At the same time, the patients receive 3-weekly trastuzumab intravenously. With the rationale of continuing a dual HER2 blockade after the failure of second-line therapy, the inclusion is limited to patients with disease progression after prior T-DM1. As of May 2020, the trial is active and recruiting, with estimated primary completion in December 2020.

NCT03525925, “Ibrutinib and Nivolumab in Treating Participants with Metastatic Solid Tumors”, is a phase I trial investigating the effect of ibrutinib and nivolumab on myeloid-derived suppressor cells. Patients with metastatic solid neoplasms and any number of prior lines of therapy are eligible for inclusion. Patients receive oral ibrutinib (420 mg daily) for 15 days and intravenous nivolumab on days 8 and 21 with subsequent repeated courses of nivolumab every 28 days until disease progression or unaccepted toxicity. The primary objective is to evaluate the levels of circulating MDSCs, with a secondary objective being the assessment of the safety of the study combination. The immunosuppressive potency of MDSCs is measured by their T cell inhibition ability and by examining antibody-dependent cell toxicity mediated by natural killer cells. The study is active but not recruiting, with estimated primary completion in September 2020.

In Table 3, we summarized the clinical trials described.

## 3. Discussion

Ibrutinib has surpassed its initial indication for hematological malignancies and is about to acquire an increasing *raison d’être* in gynecological malignancies and breast cancer. It has become clear that the mode of action extends beyond B cells, as the compound affects signaling pathways downstream of multiple other receptors [4,8]. Its promising efficacy on solid tumors can be explained by different means. As different tumor types rely on heterogeneous intracellular pathways and greatly vary in driver mutations, gene expression, and protein endowment, it is important to note that ibrutinib’s *modus operandi* in one tumor cannot be transferred to another. The mechanisms described below consist of complex interactions between the drug, the tumor, and its microenvironment.

First, ibrutinib shows on-target BTK inhibition in certain solid tumors. Ovarian cancer cells were found to express BTK, correlating with higher clinical disease stage, the risk for metastasis, and survival [42]. This could be used as both a prognostic predictor and a pharmacological target. Furthermore, Zucha et al. showed that ibrutinib diminished the self-renewal capacities and the proportion of CSCs in ovarian cancer due to their BTK dependence. Cisplatin resistance of CSCs was surmounted, promoting a synergistic effect of both drugs [42]. While the calculations performed by Di et al. implied no use as an ovarian cancer drug candidate [21], their results were purely computational and were never confirmed by testing. In vitro testing performed by Lohse et al., however, did show a weak effect on certain cell lines. Further on-target effects were identified by Eifert et al. [36] and Wang et al. [37]. BTK-C, an isoform of BTK, was expressed in 30% of breast cancer tissue, with an even higher percentage (43%) in HER2^+^ tumors. The observed reduction of HER2^+^ cell viability is proposed to be mediated via BTK’s role in the AKT-ERK axis. A synergistic effect between direct inhibition of BTK (-C) and HER2 (as described below) is likely.

Second, the effects of ibrutinib in- and outside the TEC family kinases, which are linked to the inhibition of certain additional proteins, do exist. In BTK, ibrutinib covalently binds to a cysteine residue in the ATP-binding domain. Several other enzymes contain homologous cysteine residues at analogous positions, as it is conserved among at least nine other tyrosine kinases [2]. An effect has been described for ERBB family kinases (EGFR, HER2, HER3, HER4), TEC family kinases (TEC, BTK, BMX, ITK, RLK), and others (JAK3, BLK, FGR, HCK, LCK, Yes/YES1) [8,45]. Even at low concentrations of 1 µmol/L, ibrutinib is able to inhibit 16 kinases by > 95% [46].

Changes in the expression or structure of tyrosine kinase receptors are known to be common in different human cancers and allow for pharmaceutical targeting. Ibrutinib rapidly binds to these targets, leading to reduced phosphorylation of kinases, such as ErbB1-3 [33]. This inhibition is irreversible, allowing for its typical once to twice daily dosage despite a short plasma half-life [2].

The inhibitory effect on EGFR has been specifically explored and is effective in different tumor cell lines, such as EGFR-mutant non-small-cell lung cancer (NSCLC) in mice [7]. Screening of large panels of cell lines has proved that ibrutinib suppresses the growth of cells suppressed by other EGFR/HER2 inhibitors [2]. Chen et al. did not find an effect of ibrutinib on HER2-negative cell lines during several broad screenings [2], a finding that strengthens the hypothesis of ERBB family blocking as a main mode of action.

The effect on the ERBB family, and especially HER2, becomes even more relevant as HER2 overexpression occurs in approximately 20% of breast cancers, resulting in more aggressive subtypes. In mice xenografted with HER2^+^ cells, a significant tumor growth inhibition was seen at nanomolar, clinically achievable concentrations, and is an encouraging result [2,33].

While antibodies targeting HER2 have made their way into everyday clinical practice since long, experiments in breast cancer cell lines have revealed ibrutinib to be a promising small-molecule addition to the arsenal. As it is most potent in HER2-overexpressing cell lines, these seem to be among the most promising candidates for possible future ibrutinib treatment, especially when considering the synergies described further above. This emphasizes the drug’s multifunctional ability to exert different antineoplastic mechanisms, as outlined earlier. Combining these multiple effects with additional pharmaceutical drugs, such as trastuzumab, is an auspicious approach and is currently being explored in a clinical trial [16]. Ibrutinib’s inhibition of bone marrow X-linked kinase (BMX) is another promising target but remains largely unexplored in gynecological tumors and breast cancer. BMX is overexpressed in several tumor types and promotes cell proliferation through the PI3K/AKT pathway. Its blockage could be of interest, especially when combined with PI3K/AKT inhibitors [4]. It has been shown to be up-regulated in breast and cervical cancer, enhancing cell proliferation, invasiveness, and migration. BMX expression correlates with tumor differentiation and TNM stage [47,48] and is able to increase apoptotic resistance to chemotherapeutic drugs [49].

Third, immunomodulation relating to MDSC’s and PD-(L)1 seems to play a key role when using ibrutinib alone or in combination with other therapeutics. MDSCs, which are present in the stroma of many different tumors, have been shown to cause evasion of antitumor immune responses by immunosuppression and to reduce the efficacy of immune therapy [3,8]. As they express BTK [35], the corresponding inhibitors may reduce MDSCs in vivo. Additionally, ibrutinib diminishes cytokine production, PD-L1 expression, and motility of MDSCs and other myeloid cells, which consequently affects tumor microenvironment regulation and has been linked to decreased tumor vasculature density [4]. In the murine model of Stiff et al., treatment with ibrutinib resulted in a reduced frequency of MDSCs in mammary tumors and significantly improved the efficacy of anti-PD-L1 therapy in breast cancer, possibly providing a novel strategy for enhancing immune therapies in solid malignancies [35]. Ibrutinib’s influence on circulating MDSCs and its synergistic effect with anti-PD1 targeted therapy are currently under investigation in humans. We highly anticipate the results of NCT03525925 [17] in order to improve our knowledge about the effects of this promising combination. The observed synergy of ibrutinib and immune checkpoint inhibition is in line with the work of Sagiv-Barfi et al., where the combination of ibrutinib with anti-PD-L1 agents proved efficient even in tumors (lymphomas) intrinsically insensitive to ibrutinib, with an approx. 50% cure rate in mice. In their breast cancer model, a similar effect was found despite BTK negativity and low PD-L1 expression. Hypothesizing that the synergistic effect of ibrutinib augments the effectiveness of the PD1/PD-L1 blockade by increasing antitumor T cell response [3], these data suggest a novel role of the compound as a booster of T cell therapies. An increase of CD4^+^ Th1/Th2 cells and tumor-specific antigen expression by CD8^+^ T cells has been outlined on a molecular basis [4], as elaborated in the next paragraph.

Fourth, antineoplastic effects can be explained by further modulation of adaptive immunity. One mechanism of tumor immune escape is to induce a Th2-dominant helper T cell response with antibody production instead of a Th1-dominant response. However, ibrutinib causes a shift towards the favorable Th1-based immune response and cell-mediated cytotoxicity [45,50,51], leading to increased maturation of cytotoxic T cells and depletion of immunosuppressive cytokines, such as IL-10 and TGF-β [3]. This effect is mediated by inhibiting interleukin-2-inducible T cell kinase (ITK), a TEC kinase that modulates the development and function of T cell immune response [34,45], whereas Th1 and CD8^+^ cells can rely on an escape route for maturation [4].

Ibrutinib’s performance varied greatly between animal and human studies in the trials described [39,40]. These contrasting results are multicausal. As with every experimental model, it is challenging to transfer the results to humans, adding numerous layers of complexity to a cell line or animal model. Even highly promising preclinical results had to be abandoned upon failing to show results in human trials. Nevertheless, the proposed mechanisms of action have been studied in-depth and sound plausible. The limited data available on human use are insufficient to render a complete picture of the drug’s capabilities, even more so, as the study population in the phase 1b/2 trial is heavily pretreated and suffers end-stage disease, which inevitably leads to short life expectancy and high drop-out quotas. In future research, clear inclusion criteria, ex vivo sensitivity, and genetic testing will be crucial in order to select prime candidates. With the optimal patient selection, we are optimistic that the encouraging preclinical results can be transferred to a broader clinical appliance.

As the paradigm in oncology has shifted from mere inhibition of cell division to a more holistic approach investigating the tumor and its environment, ibrutinib, with its inhibitory and immunomodulating effects, takes up an intriguing role. Future research should clearly focus on its use in HER2^+^ tumors as well as its possibilities as an enhancer of chemo- and immune therapy. In the near future, the results of the ongoing trials NCT03379428 and NCT03525925 could add further evidence in respect thereof.

## 4. Materials and Methods

### 4.1. Inclusion Criteria

For this review, studies were considered eligible if they covered (at least in one aspect) the investigation or use of ibrutinib in gynecological malignancies (including breast cancer) in theoretical or empirical (human, animal, or in vitro) studies. As per protocol, fundamental research, preclinical, and clinical studies in English published until April 1, 2020, were considered for this review. We did not include reviews, conference abstracts, or notes without an abstract. No other restrictions were used.

Clinical trials were considered eligible if they investigated the use of ibrutinib in female genital tract tumors or breast neoplasms with no further restrictions.

### 4.2. Search Strategy

We performed electronic database searches in EMBASE (DataStar Version, Cary, NC, USA) and MEDLINE (Ovid version, New York, NY, USA). The search was conducted using the following Emtree/MeSH terms: “ibrutinib” or “PCI32765” in combination with “female genital tract tumor” (Emtree) or “genital neoplasms, female” (MeSH), and “breast tumor” (Emtree) or “breast neoplasms” (MeSH). The Emtree/Mesh terms were expanded in order to include narrower terms (e.g., “ovary tumor” or “breast cancer”). Boolean operators were used as conjunctions to extend or limit the search, as described in Table A1 and Table A2. All matching publications were screened. The manual exclusion was performed after screening the titles and/or abstracts to identify eligible publications. The bibliographies of all articles that fulfilled the inclusion criteria were reviewed to identify other articles that fulfilled the inclusion criteria.

Clinical trials were searched in Clinicaltrials.gov (United States Library of Medicine, Bethesda, MD, USA) using the following terms in the respective search fields condition or disease: “solid tumor”, “gynecologic cancer”, “ovarian cancer”, “fallopian tube cancer”, “peritoneal cancer”, “cervical cancer”, “uterine cancer”, “corpus uteri cancer”, “corpus uteri carcinoma”, “endometrial cancer”, “vaginal cancer”, “vulvar cancer”, “breast cancer”; other terms (medication): “ibrutinib”.

### 4.3. Data Collection Process, Data Items

Due to inevitable methodological inconsistencies in the studies retrieved and experiments conducted, standardized data extraction could not be performed. Findings were extracted independently in a narrative way to the best of our ability.

## Figures and Tables

**Figure 1 ijms-21-04154-f001:**
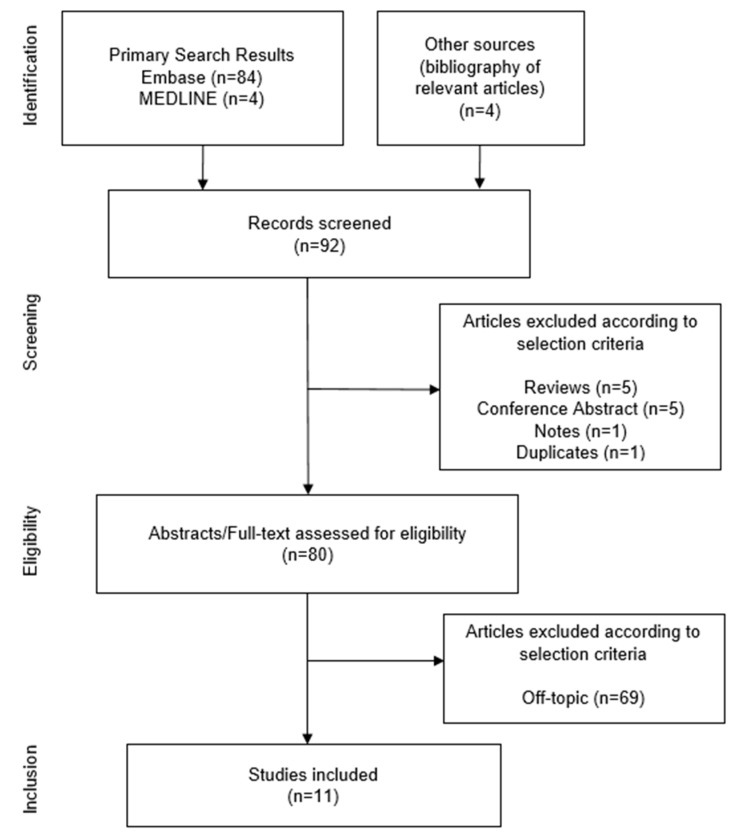
Flow diagram for the identification of relevant studies.

**Table 1 ijms-21-04154-t001:** Current evidence of the efficacy of ibrutinib in breast cancer.

First Author [Ref.]	Year	Study Type	Model	Histology	Outcome Measures	Main Outcomes
Grabinski	2014	Preclinical: in vitro	Cell culture	Breast cancer (HER2^+^)	Cell viability, phosphorylation of receptor tyrosine kinases	Significant reduction in the viability of HER2^+^ cell lines due to ibrutinib. IC50 values at nanomolar concentrations. Synergistic cell viability reduction of ibrutinib and PI3K/mTOR inhibitor dactolisib. Reduced MAPK and AKT phosphorylation.
Sagiv-Barfi	2015	Preclinical: in vitro + in vivo (animal)	Cell culture, xenograft mouse model	Breast cancer (TNBC)	Cell viability, tumor size, no. of metastases; Animal survival	Combination of ibrutinib and anti–PD-L1 inhibits growth of TNBC. The combination therapy generates specific antitumor T cells.
Chen	2016	Preclinical: in vitro + in vivo (animal)	Cell culture, xenograft mouse model	Breast cancer (Her2^+^)	Cell viability, tumor size, drug exposure	Inhibition of growth and suppression of key signaling pathways in HER2^+^ breast cancer cell lines. Clinically achievable drug levels suppress HER2^+^ human breast cancer growth in xenograft mouse models.
Stiff	2016	Preclinical: in vitro + in vivo (animal)	Cell culture, xenograft mouse model	Breast cancer (TNBC)	BTK expression, MDSCs frequency in vivo (by IHC), tumor size	Human and murine MDSCs express BTK. Ibrutinib modulates MDSCs’ cell function and generation and diminishes MDSCs in tumor-bearing mice. Ibrutinib potentially enhances immune-based therapies in solid malignancies.
Wang X	2016	Preclinical: in vitro + in vivo (mice)	Cell culture, xenograft mouse model	Breast cancer (Her2^+^)	Cell viability, tumor size	Ibrutinib is more potent in inhibiting HER2^+^ cell than lapatinib. Ibrutinib blocks EGFR, HER2, ErbB3, ErbB4 at its downstream effectors.
Di	2019	Preclinical: bioinformatics	Drug functional similarity network	Breast cancer, ovarian cancer	Prior score, false discovery rate	Ibrutinib ranks 3rd in the prioritized list of candidate breast cancer drugs and is considered to have great potential effects.
Hong	2019	Phase Ib/II	Human	Breast cancer (TNBC, HER2^+^), pancreatic cancer, NSCLC	Overall response rate, progression-free survival, overall survival	Recommended phase 2 dose: 560 mg ibrutinib daily, durvalumab 10 mg/kg i.v. q2w. ORR: 3% for breast cancer; 2% for pancreatic cancer; 0% for NSCLC. Limited antitumor activity; acceptable safety profile.
Varikuti	2020	Preclinical: in vivo (animal)	Cell culture, xenograft mouse model	Breast cancer (Luminal B)	Cell viability, tumor size, metastasis count, cell maturation analysis, T-cell proliferation and effector function	Ibrutinib inhibits tumor growth in vitro. Treated mice suffer lower tumor burden and less metastases. MDSCs switch phenotype to mature dendritic cells in vitro and less MDSCs and more DCs in vivo. Ibrutinib induces antitumor Th1 and CTL response.

Legend: q2w: every 2 weeks; BTK: Bruton’s tyrosine kinase; i.v.: intravenously; CTL: cytotoxic T lymphocytes; DC: dendritic cell; IC50: half maximal inhibitory concentration; MDSC: myeloid-derived suppressor cell; NSCLC: non-small-cell lung cancer; ORR: overall response rate; TNBC: triple-negative breast cancer.

**Table 2 ijms-21-04154-t002:** Current evidence of the efficacy of ibrutinib in gynecological malignancies.

First Author [Ref.]	Year	Study Type	Model	Histology	Outcome Measures	Main Outcomes
Zucha	2015	Pre-clinical: in vitro	Human tissue samples, cell culture, spheroids	Ovarian cancer	Cell viability, BTK expression	BTK is a histological biomarker and a prognostic predictor of ovarian cancer. Ovarian CSCs express BTK and contribute to cisplatin resistance. BTK inhibition targets CSCs and reduces their survival against cisplatin. Cisplatin–ibrutinib combination has synergistic effects in eliminating ovarian cancer cells.
Tamura	2018	Preclinical: in vitro	Cell culture, patient-derived tumor organoids	Endometrial cancer	Tumor size, inhibitory concentration	Ibrutinib inhibits the growth of carboplatin/paclitaxel-resistant cells at lower concentrations than carboplatin, paclitaxel, methotrexate, and vindesine. Less efficacy in clear-cell adenocarcinoma cell lines.
Di	2019	Preclinical: bioinformatics	Drug functional similarity network	Ovarian cancer, breast cancer	Prior score, false discovery rate	Low rank in ovarian cancer drug candidates.
Lohse	2019	Preclinical: in vitro	Drug sensitivity testing in patient-derived cell lines	Ovarian cancer	Modified drug sensitivity scoring	Weak effect on endometrioid and papillary-serous cell line. No effect on clear-cell cancer lines.

Legend: BTK: Bruton’s tyrosine kinase; CSC: cancer stem cells.

**Table 3 ijms-21-04154-t003:** Clinical trials investigating the use of ibrutinib in gynecological malignancies, including breast cancer.

Ref.	Title	Phase/ Status	Brief Summary	Primary Outcomes
NCT02403271 [15]	A Multi-Center Study of Ibrutinib in Combination with MEDI4736 [Durvalumab] in Subjects With Relapsed or Refractory Solid Tumors	Phase Ib/II; completed ^1^ (Ref.: Results section, Hong et al., 2019 [41])	A phase 1b/2, multi-center study to assess the safety and efficacy of ibrutinib in combination with durvalumab (MEDI4736) in participants with relapsed or refractory solid tumors.	Safety and tolerability, dosage, overall response rate per RECIST 1.1
NCT03379428 [16]	Trial of Ibrutinib Plus Trastuzumab in HER2-Amplified Metastatic Breast Cancer	Phase I/II; recruiting ^1^	Open-label dose-escalation study to evaluate the maximum-tolerated dose and dose-limiting side effects of daily oral ibrutinib in combination with trastuzumab i.v. q3w, in patients with HER2-amplified metastatic breast cancer with progress after prior therapy with T-DM1.	Maximum-tolerated dose, clinical benefit rate
NCT03525925 [17]	Ibrutinib and Nivolumab in Treating Participants with Metastatic Solid Tumors	Phase I; active, not recruiting ^1^	A phase I trial investigating the effect of ibrutinib and nivolumab on circulating levels of MDSCs in patients with metastatic solid tumors and assessing the safety of the study combination.	Circulating levels of myeloid-derived suppressor cells

^1^ Recruiting status as of May 1st, 2020.

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
