# Peer review of "Ibrutinib in Gynecological Malignancies and Breast Cancer: A Systematic Review"

_ijms, 2020, doi:10.3390/ijms21114154_

Round 1
Reviewer 1 Report
This is an interesting systematic review that summarizes the evidence about the use of Ibrutinib, a Bruton's tyrosine kinase inhibitor, for the treatment of gynecological neoplasms and breast cancer.
The work is exhaustive and the literature research is complete.
I suggest to clearly separate the two topics elsewhere across the manuscript.
Reviewer 2 Report
Metzler et al. present a systematic review on the use of ibrutinib in gynecological malignancies and breast cancer. It has well-structured sections, which broadly covers pre-clinical as well as clinical studies, and discusses the efficacy of ibrutinib in these settings. Overall, it will serve as a useful compilation of current information on the use of ibrutinib in these settings. I only have some minor suggestions:
1) It would help the reader, if a section on other Btk kinase inhibitors were included. Do some of these other inhibitors have broad effects similar to ibrutinib? Or does ibrutinib have a unique target kinase profile that makes it stand out?
2) Accordingly, it might be useful to state the rationale for focusing on ibrutinib. Or alternatively, briefly discuss the potential of other Btk inhibitors in these settings as well.
Reviewer 3 Report
The review by Metzler et al. (Ibrutinib in Gynecological Malignancies and Breast Cancer: A Systematic Review) summarises available experimental and human data on the effects of ibrutinib, a Bruton's tyrosine kinase in breast and gynecological cancers. The review is very brief, containing only 31 references, which may be due to limited relevant data, still, more effort could had been put in literature search. The authors included 11 relevant preclinical studies and 3 clinical studies, which were described in the manuscript and summarised in 2 tables. The draft is reasonably organized; however, it needs an improvement. I have several remarks.
- Abstract is nondescript and needs to be rewritten to provide more information on the ibrutinib efficacy.
- The mechanism of action of ibrutinib in the introduction section is poorly described, the authors must perform more thorough literature search to improve this. Reported side effects and toxicity of ibrutinib should be covered too.
- The results part is a summary of chosen relevant studies put one after another and hardly any comparison and conclusion of these studies is made in the discussion section, which is disappointing. The information on used doses should also be included.
Most importantly, regarding the limited data on ibrutinib efficacy in chosen cancers, I do believe the authors should expand the draft and summarise data available from other cancers too as systematic reviews are missing not only in relation to breast and gynecological cancers. In this case, the article could be of interest for researchers and/or clinical investigators.
Round 2
Reviewer 3 Report
No further comments.